# Rotating robots move collectively and self-organize

Christian Scholz [1,2], Michael Engel[1] & Thorsten Pöschel [1]

Biological organisms and artificial active particles self-organize into swarms and patterns. Open questions concern the design of emergent phenomena by choosing appropriate forms of activity and particle interactions. A particularly simple and versatile system are 3D-printed robots on a vibrating table that can perform self-propelled and self-spinning motion. Here we study a mixture of minimalistic clockwise and counter-clockwise rotating robots, called rotors. Our experiments show that rotors move collectively and exhibit super-diffusive interfacial motion and phase separate via spinodal decomposition. On long time scales, confinement favors symmetric demixing patterns. By mapping rotor motion on a Langevin equation with a constant driving torque and by comparison with computer simulations, we demonstrate that our macroscopic system is a form of active soft matter.

[1] Institut für Multiscale Simulation, Friedrich-Alexander-Universität Erlangen-Nürnberg, 91052 Erlangen, Germany. [2] Institut für Theoretische Physik II: Weiche Materie, Heinrich-Heine-Universität Düsseldorf, 40225 Düsseldorf, Germany. Correspondence and requests for materials should be addressed to C.S. (email: christian.scholz@fau.de)

Swarms of robots mimic collaboration of animals and can be programmed to arrange into shapes by information exchange among individuals[1]. A purely physical approach to achieve collective behavior is self-organization, where patterns emerge outside of thermal equilibrium[2]. Particularly simple examples are walking robots known as bristlebots, artificial bugs, or vibrobots. To excite motion, energy is input by a motor carried on top of the robot vibrating its body[3] or, alternatively, by a vibrating baseplate[4, 5]. Each robot consists of a rigid body and tilted elastic legs. The legs act as elastic springs and convert momentum of the baseplate into motion of the robot at each impact[6–9]. In general, parallel legs apply translational forces (walker), while legs arranged in a circular manner apply torques (rotor, spinner or vibrot). As a result, these robots move, similar to mechanically vibrated polar disks[10, 11]. Together with biological[12] and synthetic microswimmers[13], they belong to the class of active soft matter. Collectives of such particles exhibit phase separation into dense clusters and a surrounding gas phase[14–19] or self-organize into swarms and flocks[3, 10, 11].

In this work, we focus on 3D-printed rotors of gear-like shape, which are expected to self-organize in ways not possible in the absence of energy input[20–25]. Existing experimental realizations of rotors depend on the application of electromagnetic fields[20–22]. This approach does not allow rotation control of individual rotors because the fields act equally on all particles. As an alternative, we use 3D printing to fabricate rotors with propulsion mechanisms that can be varied for each particle individually. We demonstrate that our setup is suitable for observing phase separation driven by active rotation[26, 27], a phenomenon reminiscent of spinodal decomposition in binary fluids with a high symmetry between both phases. Our experiment is a simple yet non-trivial example for chiral symmetry breaking in a classical system, far beyond the well-known swarming and flocking behavior of active agents.

## Results

**Phase separation of opposite rotors in experiment**. A mixture of 210 clockwise and 210 counter-clockwise spinning rotors (Fig. 1a, b) with area filling fraction of 52% is prepared by placing the particles in a fully mixed checkerboard configuration (Fig. 1c). Once the vibration is turned on, the initial state quickly decays into a disordered configuration (Fig. 2a, Supplementary Fig. 1, and Supplementary Movie 1). After about 60 s, domains of equally rotating particles are visible and then grow until the system is almost fully segregated (Fig. 2b, c). During the segregation process, we observe super-diffusive motion along the interface, also known as edge currents (Fig. 2g)[26, 28].

Particle segregation is robust in a wide range of packing fractions $\phi$. To show this, we vary the density of a smaller system (enclosing ring 300 mm diameter) in the range $\phi = 0.3–0.7$, where the system is stable against gravitational drift and still below the transition to crystallization (Fig. 2h). We start at $\phi = 0.3$ and successively add particles and wait for the system to relax (1–2 h per step). After reaching $\phi = 0.7$, we reverse the process by removing particles to test the reversibility of phase separation. Domains of likewise spinning rotors form at packing fractions up to $\phi = 0.6$. To demonstrate the efficiency of the demixing process, we calculate the relative difference of equal and opposite nearest neighbars $(N_{op} - N_{eq}) / N_{tot}$ for each packing fraction, which in the absence of system boundaries is 0 for the fully mixed state and 1 for the fully demixed state. A plateau is found in the range $\phi \lesssim 0.55$ before the system starts mixing again at higher packing fraction (Fig. 2i).

Phase separation is counter-intuitive at first glance because particles interact purely repulsively and the spinning motion of equal rotors is blocked when in contact (Fig. 3a–d). In contrast,

opposite rotors come apart more easily after collision (Fig. 3e–g). As a result of the longer contact time of equal rotors, the system exhibits spontaneous separation[26]. We observe that the segregation efficiency decreases for high packing fraction (Fig. 2i), possibly because there is less freedom for likewise spinning particles to perform collective rotation. In the following experiments and simulations, we fix the packing fraction again to 52% by using the large enclosing ring with 420 particles. At this packing fraction pronounced segregation is generally observed.

We quantify edge currents by measuring the mean squared displacement (MSD) of particles moving close to the interface between domains of opposite rotors. MSD is recorded after a fixed time interval in dependence on the distance to the interface (Fig. 4a) as well as at fixed initial distance to the interface in dependence on time (Fig. 4b). We observe that particles close to the interface are more mobile and for distances greater than three particle diameters the displacement is independent of the distance. Furthermore, the time dependence of the MSD is quadratic at the interface and linear at larger distances.

**Mapping of single-particle motion to Langevin dynamics**. Numerous results in the field of active matter have been discovered in simulations using effective equations of motion, typically with colloidal applications in mind in which active motion is driven by chemical gradients[29]. In our system, particle rotation is the result of a purely mechanical ratcheting mechanism[5, 8]. Interactions are extremely short-ranged, collisions dissipative, and hydrodynamics negligible. We test whether our macroscopic experiment agrees with the approach in ref.[26] and map it to a system of coupled Langevin equations with a driving torque of constant magnitude.

The equations of motion for the $i$-th particle with velocity $\mathbf{v}_i$ and angular momentum $\omega_i$ read

$$M\frac{\partial \mathbf{v}_i}{\partial t} = \mathbf{F}_i + \gamma^{\mathrm{T}}\left(\sqrt{2D^{\mathrm{T}}}\boldsymbol{\eta}_i(t) - \mathbf{v}_i\right), \qquad (1)$$

$$I\frac{\partial \omega_i}{\partial t} = \tau_i + \gamma^{\mathrm{R}}\left(\sqrt{2D^{\mathrm{R}}}\zeta_i(t) - \omega_i\right) + \tau_i^{\mathrm{D}}, \qquad (2)$$

where $M$ and $I$ are mass and moment of inertia of a single rotor,

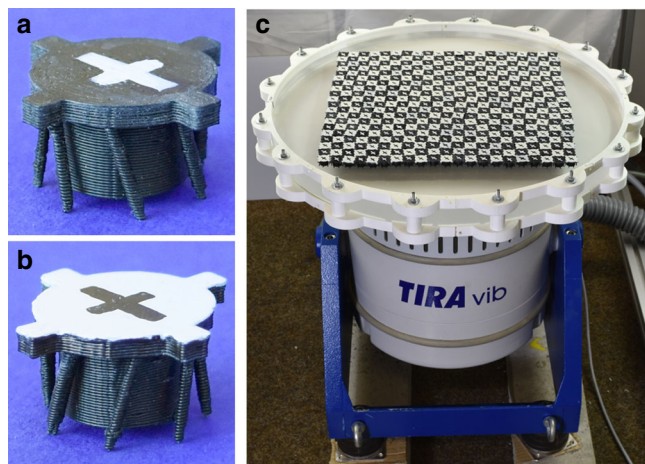

**Fig. 1** 3D-printed spinning robots and experimental setup. **a** Clockwise and **b** counter-clockwise spinning rotors (diameter σ = 15 mm). Four satellites attached to the edge of the body enhance steric interactions between particles. **c** Electromagnetic shaker with circular acrylic baseplate and smooth circular enclosing ring (diameter 470 mm). Initially, the rotors are placed in a checkerboard configuration with alternating clockwise and counter-clockwise spinning rotors

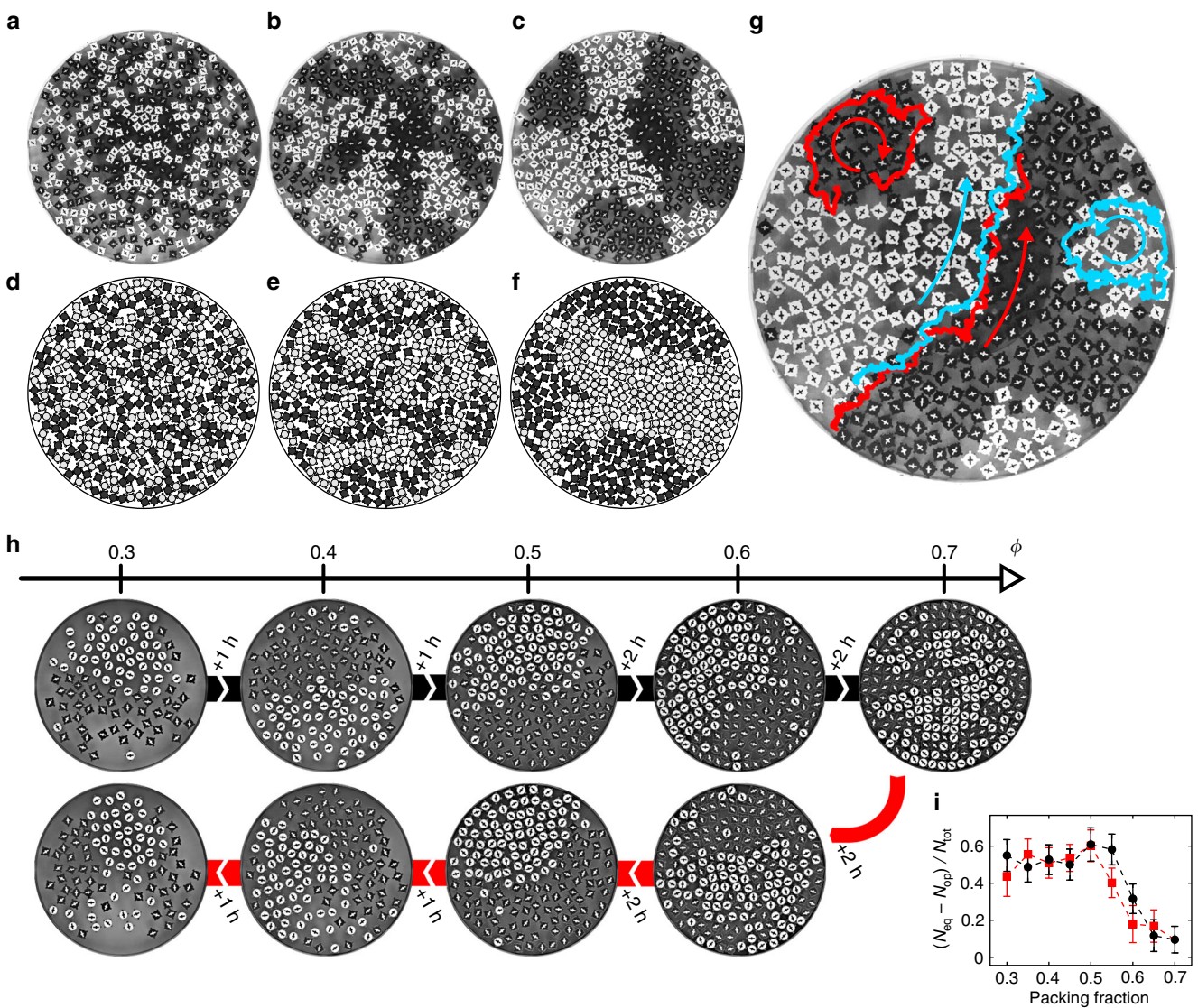

**Fig. 2** Phase separation of a binary mixture of rotors. Experimental system at times **a** 10 s, **b** 60 s, **c** 900 s after initialization. **d**–**f** Snapshots of computer simulations at the same times. **g** Rotors at the interface perform super-diffusive motion along the interface. Representative trajectories are shown for clockwise transport of clockwise spinning particles (black particles, red trajectories) and counter-clockwise transport of counter-clockwise spinning particles (white particles, blue trajectories). Arrows indicate the direction of motion. For further visualizations of edge currents see Supplementary Fig. 2–3 and Supplementary Movies 2, 3. **h** Snapshots of the system at different packing fractions after successively adding and then removing particles. Segregation occurs in a wide range of packing fractions of $\phi = 0.3$–$0.6$ and breaks down at larger $\phi$. **i** Relative difference of equal and opposite nearest neighbors $(N_{\text{eq}} - N_{\text{op}}) / N_{\text{tot}}$ during a time interval of 20 min at fixed $\phi$. Error bars are defined as standard deviation. For $0.3 \leq \phi \lesssim 0.55$, the relative difference is independent of packing fraction indicated by a plateau

$\gamma^{\text{T}}$ and $\gamma^{\text{R}}$ are translational and rotational damping coefficients, respectively, $\tau_i^{\text{D}} = \pm$ const is a driving torque, and $\mathbf{F}_i$ and $\tau_i$ are forces and torques resulting from collisions with other particles and the enclosure. The random forces and torques are uncorrelated Gaussian processes with $\langle \boldsymbol{\eta}_i^{(m)}(t)\boldsymbol{\eta}_j^{(n)}(t') \rangle = \delta_{ij}\delta(t - t')\delta_{mn}$, where $m,n \in \{x,y\}$ and $\langle \zeta_i(t)\zeta_j(t') \rangle = \delta_{ij}\delta(t - t')$. In the model, their amplitudes are functions of the diffusion coefficients $D^{\text{T}}$ and $D^{\text{R}}$. However, in contrast to colloidal systems, the random forces do not describe thermal noise but an instability that arises from the vibration mechanics[8]. This means diffusion and damping constants are not related through the Stokes–Einstein relation.

We measure the mass $M$ and calculate $I$ from particle shape and density. All other parameters are extracted from vibration experiments. The diffusion coefficients $D^{\text{T}}$ and $D^{\text{R}}$ are obtained from stochastic velocity fluctuations of a single-particle trajectory

(Fig. 5a, b) by fitting the slope of the MSD (Fig. 5c). The driving torque is given by $|\tau^{\text{D}}| = \gamma^{\text{R}}\overline{\omega}$ in the steady state. It is only weakly dependent on the vibration amplitude $A$ for $A > 70\,\mu\text{m}$, where we observe an average angular velocity $\overline{\omega} = \frac{1}{T}\int_0^T \omega(t)\mathrm{d}t \sim 10\,\text{rads}^{-1}$ (Fig. 5d). The translational damping coefficient $\gamma^{\text{T}}$ is obtained from the response of an isolated rotor to an instantaneous kick. A rotor is perturbed and the resulting velocity decay measured (Fig. 5e). We find a good fit with an exponential decay, $v \propto \exp(-t\gamma^{\text{T}}/M)$, similar to a particle suspended in a viscous fluid, which confirms that the assumption of a velocity-dependent frictional force in the Langevin equations (1) and (2) is justified. Finally, the rotational damping coefficient $\gamma^{\text{R}}$ is determined from the acceleration of a stopped rotor due to the driving torque (Fig. 5f). Again, we find a good fit with an exponential function, $\overline{\omega} - \omega \propto \exp(-t\gamma^{\text{R}}/I)$.

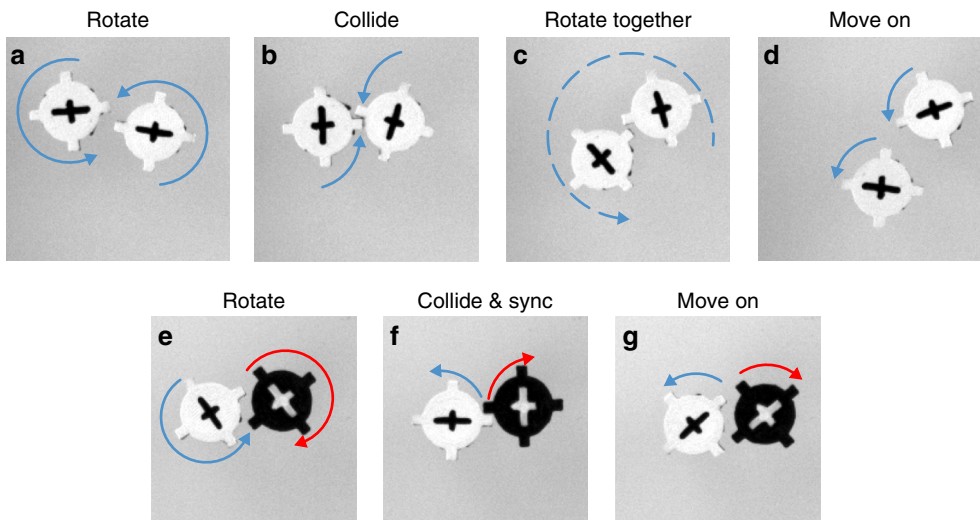

**Fig. 3** Typical collision between equal and opposite rotors. Collision sequence for likewise spinning rotors is **a** individual rotation, **b** collision, **c** collective rotation and **d** individual motion (duration between frames is 0.12 s). When **e** opposite rotors collide, we observe **f** synchronization followed again by **g** individual motion (duration between frames is 0.07 s). Due to the additional collective rotation likewise spinning rotors spend more time together, which causes a net attraction

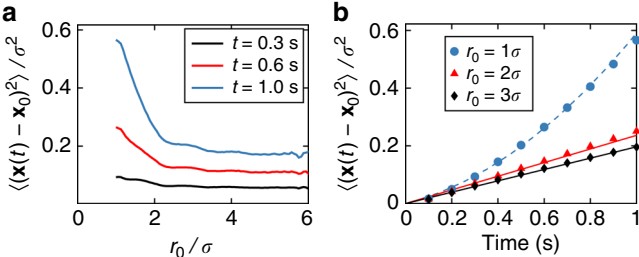

**Fig. 4** Mean squared displacement (MSD) near the interface between domains of opposite rotors. **a** MSD after 0.3 s, 0.6 s, and 1.0 s in dependence on the distance to the closest opposite rotor $r_0$ (i.e., the interface) at time $t = 0$ in units of particle diameter σ. **b** Time dependence of the MSD for particles at distances 1σ, 2σ and 3σ from the interface. MSD of particles in direct contact with the interface increases quadratically (dashed line). Particles further away from the interface show a linear MSD (solid lines)

The validity of the equations of motion and the parameters extracted from experiment (Table 1) are tested with Langevin dynamics simulations. A typical simulation evolution is shown in Fig. 2d–f (Supplementary Fig. 4 and Supplementary Movie 4). Visual inspection shows good agreement with the experiment.

**Collective behavior on long time scales**. We compare the domain size evolution of seven experimental runs and 128 Langevin dynamics simulations for a duration of 1000 s (Fig. 6a). Variations among experimental runs appear to be slightly larger than variations among the simulation runs, possibly caused by mass polydispersity and shape imperfections of our 3D-printed rotors. Still, overall good agreement in the average coarsening behavior is observed. This demonstrates that the system of Langevin equations fitted to single-particle behavior also reproduces collective dynamics well.

Dimensional analysis and numerical solutions of the Cahn–Hilliard model for spinodal decomposition predict that domain size growth follows $s(t) = t^{1/3}g$[30, 31] with growth coefficient $g$ on the order of $(Dw)^{1/3}$[32, 33]. Here, $D$ is the particle self-diffusion coefficient in the dense rotor system and $w$ is the width of the transition zone between domains. The self-diffusion coefficient is approximated from experiment and simulation as $D = 2 \times 10^{-5}$ m² s⁻¹ and the transition zone width is assumed to be about one particle diameter, $w = 15$ mm. Thus, $g$ should be on the order of $10^{-2}$ m s⁻¹/³. Our data is consistent with $g = 1.5(5) \times 10^{-2}$ m s⁻¹/³ (thick black lines in Fig. 6a). Nevertheless, the comparison to a power-law is only qualitative and is affected by the confinement condition. Simulations of large systems of rotors without confinement have already shown with high accuracy that the Langevin model used here leads to spinodal decomposition with exponent 1/3[26]. Larger setups than ours are necessary to accurately determine the growth exponent in experiment.

Because random forces and the viscous term in the contact force are frequently neglected in the literature, but could be relevant in our experiment, we test simulations with $D^T = D^R = t_{vis} = 0$ (Fig. 6b). The deviation of these simulations from the ones with the full model is small, justifying the simplification. We use this simplification to reduce computational cost and study a large ensemble of realizations over longer times than attainable from experiment in a reasonable time.

At longer times, confinement affects the kinetics of phase separation strongly[28, 34]. Clear deviations from a power law occur when we extend the observation time to 10000 s (Fig. 6c). The system remains trapped in metastable states for a certain time period, a phenomenon that is apparent as plateaus in the domain size growth plots, before coarsening eventually saturates in the completely phase-separated state. From the analysis of snapshots, we discover that the plateaus correspond to the following high-symmetry states: the completely phase-separated state (Fig. 6d), a symmetric state with three domains (Fig. 6e) and two types of metastable states with four domains (Fig. 6f, g). Figure 6h–i show corresponding high-symmetry states observed in experiment. Domain interfaces in the symmetric states are curved such that they intersect the boundary at angles close to 90° (dashed red lines in Fig. 6i). Domains with more straight interfaces are harder to break up because the interfacial motion exerts a lower pressure, which means the system remains in these metastable state for longer time. Surprisingly, this observation is reminiscent of phase segregation in immiscible fluids under strong confinement[35, 36], even though all essential preconditions applicable to molecular fluids are not fulfilled in our system. In particular, pressure is generally not a state variable in active systems[37].

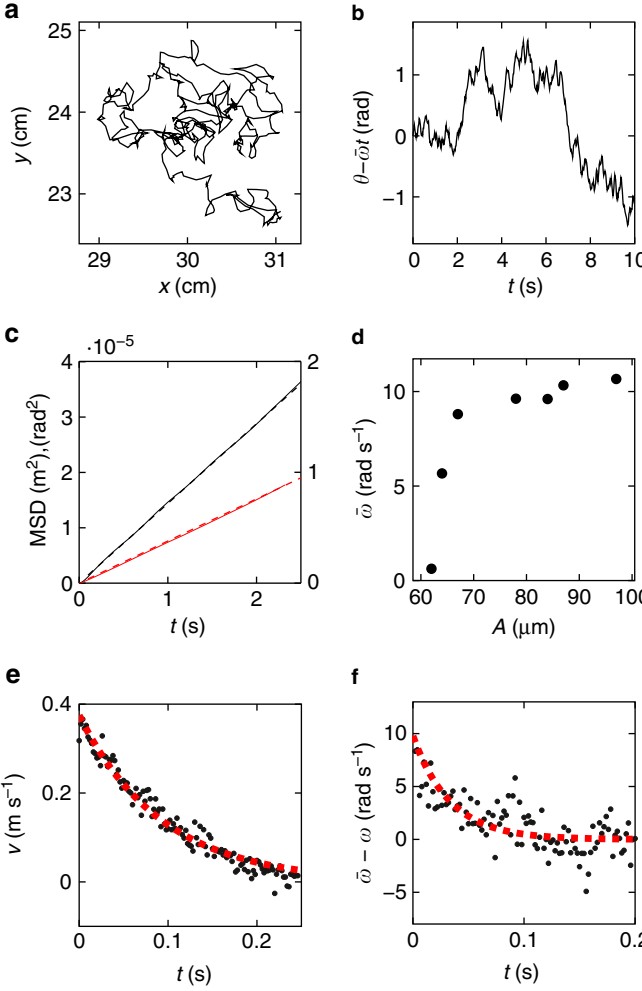

**Fig. 5** Dynamic properties of a single rotor. **a** Example trajectory of a rotor on the vibrating table. **b** Particle orientation in the co-rotating frame of reference $\theta - \overline{\omega}t$. **c** Mean squared displacement of translational (red) and rotational (black) degrees of freedom. **d** Average angular velocity $\overline{\omega}$ at constant frequency $f = 80$ Hz in dependence on the excitation amplitude $A$. **e** Relaxation of translational velocity after a kick. **f** Relaxation of angular velocity after releasing a particle. Fits with linear functions (**c**) and exponentials (**e**, **f**) are shown with dashed curves

**Table 1 Simulation parameters obtained from experiment**

| Parameter | Symbol | Value |
|---|---|---|
| Particle mass | $M$ | $0.969(5) \times 10^{-3}$ kg |
| Moment of inertia | $I$ | $2.21(1) \times 10^{-8}$ kg m$^2$ |
| Translational diffusion | $D^T$ | $7.2(2) \times 10^{-6}$ m$^2$ s$^{-1}$ |
| Rotational diffusion | $D^R$ | $0.19(3)$ rad$^2$ s$^{-1}$ |
| Driving torque | $\tau^D$ | $\pm 6.7(29) \times 10^{-6}$ kg m$^2$ s$^{-2}$ |
| Translational damping | $\gamma^T$ | $0.010(2)$ kg s$^{-1}$ |
| Rotational damping | $\gamma^R$ | $7(3) \times 10^{-7}$ kg m$^2$ s$^{-1}$ |

While the simulation typically reaches a fully demixed symmetric steady state, the experimental system is more sensitive and the fully segregated symmetrical state is found only for small system sizes (Fig. 2h). For the large system of Fig. 6, the time to reach the final state is so long that gravitational drift eventually leads to asymmetry and we only observe nearly full demixing. Despite our attempts to correct the tilt down to 0.001 degree and

careful re-adjustment before each run, tilt is gradually introduced due to the constant vibrations of the system restricting the maximal experimental time. Edge currents at the boundary appear to prevent the achievement of the symmetric fully segregated state (Fig. 7).

## Discussion

Our findings suggest an analogy of our granular experiment and the behavior of microscopic soft matter systems. Past simulations predicted that phase separation of rotors is still possible in the presence of hydrodynamic interactions[27]. We therefore expect that the evolution observed in our experiments could agree with that of smaller active rotors, like colloidal and molecular spinners in liquid crystals or biological motors[38–43], even though the confirmation of this phenomenon remains an ongoing experimental challenge. Additionally, our joined experimental and numeric approach strongly suggests that analytic descriptions of self-organization for microscopic systems are also applicable to active matter on large length scales. In future, effects characteristic for binary fluids, such as emulsification or critical Casimir forces, could be studied on the macroscopic scale with active rotors or other 3D-printed vibrated particles.

## Methods

**Particle fabrication**. The rotor geometry consists of two connected central cylinders, a cap (diameter 15 mm, height 2 mm) and a body (diameter 11 mm, height 6 mm). Four satellites (sidelength 3 mm) and seven tilted legs (length 8.5 mm, diameter 1 mm, angle 18degrees) are attached to the cap. Rotors are manufactured by 3D printing using fused deposition modeling from acrylonitrile butadiene styrene (ABS) for experiments on the large plate and stereolithography using acrylate-based photopolymer for experiments on the small plate. To simplify the detection of positions and orientations from video recordings, we mark the rotors with a cross at the top (either painted or with a printed label). To avoid alternative segregation mechanisms commonly observed in granular systems, such as size-dependent or weight-dependent segregation, the particles must be manufactured with sufficiently high accuracy. We also need to consider other effects that might lead to additional inaccuracy, like positioning on the build plate (due to misalignment, particles close to the edges of the build plate can slightly differ in height) and the role of secondary manufacturing steps (painting the cross at the top). The 3D printing fabrication via fused deposition modeling employed in this work achieves a maximum accuracy of 0.1 mm. Overall, 300 particles of each species are fabricated. Particles with a mass that deviates more than 5% from the average mass (Fig. 8) were successively discarded until 210 particles of each species remained. The resulting distribution has a standard deviation of <2%. The mean weight of each species deviates by <0.5%. Particles created by stereolithography, which achieves a higher accuracy, do not require post-processing.

**Vibrating table**. The active rotors are excited by vertical vibrations of circular acrylic baseplates (diameter 300 and 480 mm, height 15 and 30 mm) attached to an electromagnetic shaker. The tilt of the plate is adjusted with an accuracy of $10^{-3}$ degrees. Motion is restricted by an enclosing ring (diameter 470 mm). The shaker is attached to heavy concrete blocks to suppress resonances. Production experiments are performed with vibration frequency $f = 80$ Hz and vibration amplitude $A = 84$ (2) μm, at which stable excitation is guaranteed. Experiments are recorded using a high-speed camera system at up to 500 frames per second with spatial resolution of $1024 \times 1024$ pixels. The rotors are tracked using standard image recognition methods with sub-pixel accuracy, which lies below the manufacturing tolerance of the 3d printer of 0.1 mm.

**Damping constants**. To measure the damping constants $\gamma^T$ and $\gamma^R$, we assume that, on average, particle motion (translation and rotation) relaxes according to Newton's equations of motion with a viscous damping term and external torque $\tau$,

$$M\dot{\mathbf{v}} = -\gamma^T \mathbf{v}, \tag{3}$$

$$I\dot{\omega} = \tau - \gamma^R \omega. \tag{4}$$

These equations are solved by an exponential with relaxation times $t^T = M/\gamma^T$ and $t^R = I/\gamma^R$.

**Moment of inertia**. It is difficult to measure the moment of inertia $I$ directly. However, since we know the mass and shape of the particle, $I$ can be calculated

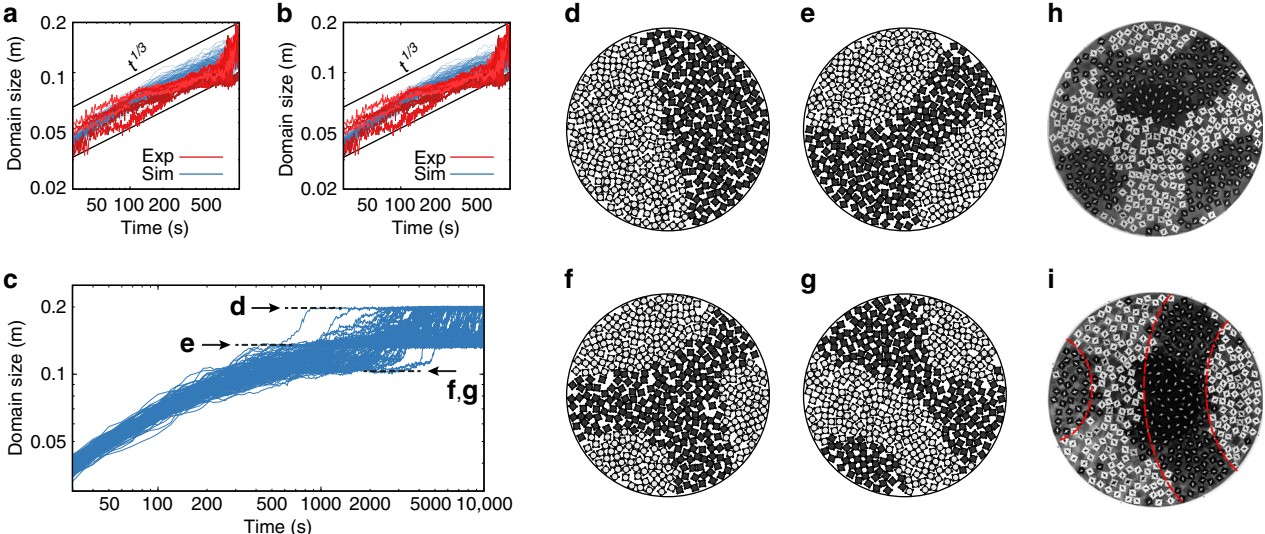

**Fig. 6** Evolution of the domain size during phase separation. **a**, **b** Domain size growth at intermediate times when the effect of confinement is weak. The data for seven experiments (red) and 128 simulations (blue) using parameters from Table 1 (**a**) and with $D^T = D^R = t_{vis} = 0$ (**b**) are shown. The evolution is compatible with a power-law $t^{1/3}$. **c** At longer times, domain size growth develops plateaus, which correspond to high-symmetry metastable states. **d**–**g** We typically observe the final Janus state (**d**), a three-domain state with mirror symmetry (**e**), and four-domain states with three-fold rotational symmetry (**f**) and a stripe geometry (**g**). **h**, **i** Symmetric states from the experiment after 690 s and 960 s, respectively, with nearly three-fold and twofold symmetry. Interfaces (dashed lines) tend to intersect the boundary at angles close to 90°. Corresponding domain sizes are indicated with arrows in **c**. For further visualizations of the development of metastable states during phase separation, see Supplementary Fig. 5–6 and Supplementary Movies 5, 6

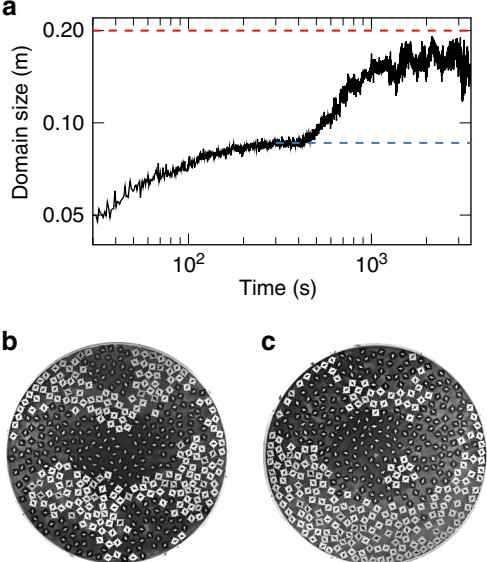

**Fig. 7** Evolution of the steady state on the large plate. **a** Long-time evolution of demixing in the large system reaches a plateau but is sensitive to currents at the boundary. **b** Metastable intermediate state after 300 s (blue dashed line in **a**). **c** The steady state after 2000 s deviates from the fully symmetric segregation (red dashed line in **a**) due to gravitational drift

from shape. The particle consists of two central cylinders (c1 and c2 with mass $m_{c1}$, $_{c2}$ and radius $r_{c1,c2}$), four cubes (satellites, s with mass $m_s$, edge length $a_s$ and distance $d_s$) on the outside and seven cylinders (legs l with mass $m_l$, radius $r_l$, length $l_l$, distance from the center $d_l$ and inclination angle $\theta$ with respect to the vertical). The total moment of inertia is the sum of individual moments considering off-central rotation for the satellites and the legs,

$$
\begin{aligned}
I &= I_{c1} + I_{c2} + 4I_s + 7I_l \\
&= \tfrac{1}{2}m_{c1}r_{c1}^2 + \tfrac{1}{2}m_{c2}r_{c2}^2 + 4\left(\tfrac{1}{6}m_s a_s^2 + m_s d_s^2\right) \\
&\quad + 7\left(m_l d_l^2 + \tfrac{1}{2}m_l(\tfrac{1}{2}r_l^2 + \tfrac{1}{6}l_l^2)\sin^2\theta + \tfrac{1}{2}m_l r_l^2\cos^2\theta\right).
\end{aligned}
\tag{5}
$$

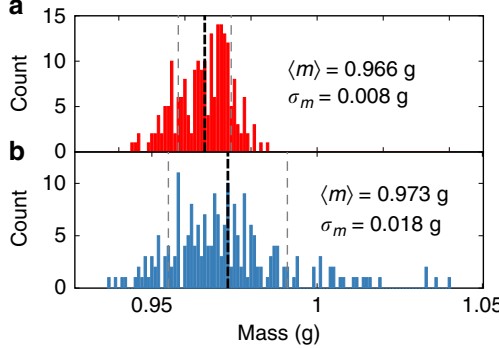

**Fig. 8** Distribution of particle masses. **a** Clockwise and **b** counter-clockwise spinning rotors mass distributions. The dashed lines indicate mean $\langle m \rangle$ and standard deviation $\sigma_m$. The width of the distribution is larger for counter-clockwise spinning rotors due to the additional manufacturing step of coloring its top face

**Langevin dynamics simulations.** Rotors are modeled by a large central disk (diameter 15 mm) and four satellite disks (diameter 3.4 mm). The diameter of the satellite disks is chosen such that the area covered by a particle equals that of the experiment. As typical for granular matter, we assume a viscoelastic normal contact force of the form $F_{vis} = F_{el}(\xi) + t_{vis}\dot{\xi}\frac{\partial F_{el}}{\partial\xi}$, where $\xi$ is the compression or indentation depth of neighboring particles (which is only microscopic in our system) and $t_{vis}$ is the viscous relaxation time describing the dissipative part of the force. Tangential forces between colliding discs are neglected. The elastic contact force of contacting cylinders is given by

$$
F_{el} = -\frac{\pi Y h \xi}{\ln \frac{e^{1+\nu}\xi}{4R_{eff}}}
\tag{6}
$$

with $R_{eff} = 2(R_1 R_2)/(R_1 + R_2)$[44] and $\xi = R_1 + R_2 - d$. Here, $Y = 2.3$ GPa is the Young modulus of the 3D printing material (ABS), $\nu = 0.35$ is the Poisson ratio, $h = 2$ mm is the cylinder height and $R_1$, $R_2$ are the cylinder radii, either 1.7 or 7.5 mm, depending on the colliding parts and $d$ is the distance between particle centers. We either set a small value of $t_{vis} = 10^{-5}$ s as typical for stiff materials or $t_{vis} = 0$ s for purely elastic collisions. Langevin dynamics simulations are performed using custom code that is included as part of the Supplementary Information.

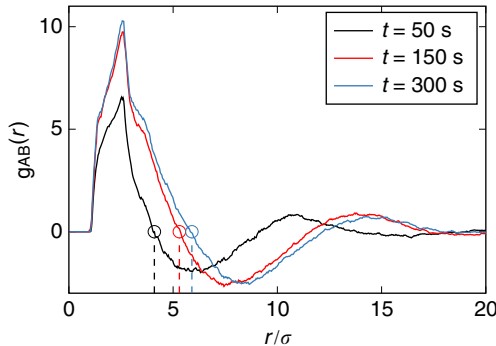

**Fig. 9** Relative radial distribution function. Radial dependence of $g_{AB}(r)$ at times $t = 50$ s, 150 s and 300 s. The first non-trivial root (marked by circles) is a quantifier for the characteristic domain size. The distance $r$ is given in units of the particle diameter $\sigma$

**Characteristic domain size**. For two particle species A and B with particle densities

$$\rho_A(\mathbf{r}) = \sum_{i=1}^{N_A} \delta(\mathbf{r} - \mathbf{r}_i), \quad \rho_B(\mathbf{r}) = \sum_{i=1}^{N_B} \delta(\mathbf{r} - \mathbf{r}_i) \qquad (7)$$

and the difference $\rho_{AB}(\mathbf{r}) = \rho_A(\mathbf{r}) - \rho_B(\mathbf{r})$, we define the relative radial distribution function for the binary mixture as the autocorrelation function

$$
\begin{aligned}
g_{AB}(\mathbf{r}_1, \mathbf{r}_2) &= \langle \rho_{AB}(\mathbf{r}_1) \rho_{AB}(\mathbf{r}_2) \rangle \\
&= +\langle \rho_A(\mathbf{r}_1) \rho_A(\mathbf{r}_2) \rangle + \langle \rho_B(\mathbf{r}_1) \rho_B(\mathbf{r}_2) \rangle \\
&\quad -\langle \rho_A(\mathbf{r}_1) \rho_B(\mathbf{r}_2) \rangle - \langle \rho_B(\mathbf{r}_1) \rho_A(\mathbf{r}_2) \rangle.
\end{aligned}
\qquad (8)
$$

If the system is homogeneous and isotropic, this expression reduces to the relative radial distribution function $g_{AB}(r)$ that only depends on the distance $r = |\mathbf{r}_1 - \mathbf{r}_2|$. It is positive if at distance $r$ mainly particles from equal species contribute to the average and negative if mainly particles of the opposite species contribute to the average. The roots of $g_{AB}(r)$ are found at distances, where particle species A and B appear with equal probability. A characteristic domain size is defined from the first non-trivial root[26, 32, 33]. Measured examples at different times are shown in Fig. 9.

**Code availability**. The Langevin dynamics simulation source code that was used to generate all simulation data for this study is included as Supplementary Information. An OpenSCAD and STL file of the rotor design is included as Supplementary Information.

**Data availability**. The data that support the plots within this paper and other findings of this study are available from the corresponding author upon request.

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

## Acknowledgements

We acknowledge funding by Deutsche Forschungsgemeinschaft through the Cluster of Excellence Engineering of Advanced Materials (EXC 315/2) and through the grant SCHO 1700/1-1. Support from the Central Institute for Scientific Computing (ZISC), the Interdisciplinary Center for Functional Particle Systems (IZ-FPS) and compute resources provided by the Erlangen Regional Computing Center (RRZE) at FAU Erlangen-Nürnberg are gratefully acknowledged.

## Author contributions

C.S. and T.P. designed the experimental setup. C.S. carried out and analyzed the experiments. M.E. and C.S. wrote the simulation code. M.E. performed and analyzed the simulations. All authors discussed the results and wrote the manuscript.

## Additional information

**Competing interests:** The authors declare no competing financial interests.

