## [Peer Review File · Nature Communications]

Reviewers' comments:

Reviewer #1 (Remarks to the Author):

This is an interesting work on segregation/coarsening dynamics of a mixture of robots rotating at different directions. The results evidence analogies with microscopic systems supposing a nice link among active matter at different length scales.

The manuscript is well written and supposes a step forward in a field that is raising considerable interest among physicists during the last years. Also, the matching of experimental results with the Langevin simulations reveals that the later can successfully reproduce macroscopic behavior. For all these reasons I think that this work could eventually deserve publication in Nature Communications.

Nevertheless, I expect something more from a paper in this journal. In particular I think that a systematic analysis of the effect of area filling fraction in the developed patterns (similar to that in ref. 4) is necessary to give the paper the robustness and generality that a journal such as Nature Communications requires.

Other mostly minor issues:

- I cannot see the supplementary videos 4, 5 and 6.
- Fig 1a and 1b are irrelevant for this work.
- A typical simulation trajectory is shown  A typical simulation evolution is shown
- The computation of the average size of domains should be explained with more detail in the Methods section.

Reviewer #2 (Remarks to the Author):

'Rotating Robots Move Collectively and Self-Organize' by Scholz et al., presents a macroscopic active matter system consisting of an ensemble of 3D-printed robots that rotate and interact sterically. They observe phase-separation into clockwise and anti-clockwise rotators, and characterize the coarsening dynamics of their system and the dynamics of individual robots in the ensemble. Their

experimental results are backed up with simulations that quantitatively reproduce the observed dynamics.

This work presents a new example of an active matter system. There are some exciting experimental results, e.g., the dynamics similar to spinodal decomposition and the multiple plateaus at longer times. However, in my opinion, the work should go further in using these results to give insights into non-equilibrium phenomena in general. Without this, the work is unlikely to be of wide general interest to the active-matter community. Thus I believe that, in its current form, this work is more suitable for a specialist journal rather than Nature Communications. With further work it could be suitable for Nature Communications. I expand on this in the following paragraphs.

As the authors state, motility-induced liquid-gas phase separation has already been observed with macroscopic robots [2], [11-13]. The authors here observe a different type of phase separation based on rotor chirality. However, I do not believe that this represents a significant advance over the previous work. If it does, the authors should explain why.

The authors claim that the early dynamics of their system is well described by spinodal decomposition. In my opinion, the experimental evidence to back up this claim is not sufficient, consisting of three experimental curves, cropped to only show a particular time window, Fig. 4a-b. One of these curves does not even show the relevant $t^{1/3}$ behaviour, as discussed below. In addition, even if this claim is true, similar results have already been found in simulations of colloidal active matter, and the authors may want to refer to these results, e.g., Stenhammar et al., PRL 111, 145702 (2013). The simulation results on long-time plateaus are interesting, but I could not tell whether the authors have any experimental results on this. Even just from the simulations, it would have been helpful to have some more discussion of these plateaus, e.g., the metastability of the associated states and the lifetime of the plateaus, and how these plateaus are expected to depend on system size (I expect that they may just be due to the small system employed, and for large-scale systems would disappear entirely).

The authors several times suggest other comparisons between their results and previous work on microscopic active and passive systems. However, the details of this comparison are often unclear, e.g., in the concluding paragraph, to what extent do the authors expect that the evolution observed in their experiments agrees with analogous results for microscopic systems, and do the references cited for this statement confirm their expectations? In general, more engagement with previous work on microscopic active matter would help to widen the interest of this work, e.g., 'Numerous results in the field of active matter have been discovered...' could do with some examples from the literature.

More detailed comments follow:

The form of F_i and τ_i should be in the main text, and this should be much clearer and more explicit even if in the SI: SI Eq. 1 presents the force as a scalar quantity, so it is unclear to me whether there is any tangential interaction between colliding cylinders, or whether the interaction is purely along the line of centres; how does the force depend on the overlap between cylinders, and what is ξ ? Also, this model employs a finite Young's modulus, yet no deformation is visible in the movies. Would a strictly rigid model, as employed in [4] give the same phenomenology, or does the modulus have a significant effect?

The definition of the random forces and torques is confusing after Eq. 1-2. I believe that this is a 2d model, so that there should only be two spatial dimensions for the random force and 1 spatial dimension for the random torque. Instead, both of these have three dimensions specified.

In Fig. 2g, arrows on the red/blue trajectories would be helpful. Also, the caption currently reads 'clockwise diffusion of clockwise-rotating particles...' etc., but the motion is non-diffusive.

The units of g should be $\text{ms}^{-1/3}$ not $\text{ms}^{1/3}$ and the parameter w should be on the order of one particle diameter, not just greater than one particle diameter, for the order-of-magnitude calculation of g to make sense.

Some explicit cross-reference to the SI would be useful, e.g., when discussing the mass and moment of inertia.

One of the red curves in Fig. 4a-b appears to show a plateau at longer times, but the authors claim that all these curves are in the intermediate-time regime where no plateaus should occur. Are these plateaus a significant, reproducible feature? More experiments are needed to demonstrate this one way or the other. Why do the authors not show the long-time dynamics courses for the experiments as well as for the simulations?

In what way is the phase separation reminiscent of droplet formation, and why are these connections surprising? A reference to a work on droplet formation would be helpful here.

Reviewer #3 (Remarks to the Author):

The authors report the first experimental investigation of the collective dynamics of spinning active particles, a subject which has recently triggered a surge of research in active-matter physics. Using suitably designed grains shaken by a vibrating stage, they demonstrate the segregation of particles spinning in opposite direction and claim the observation of subdiffusive motion at the domain interfaces. These observations are consistently reproduced by numerical simulations similar to that first introduced in [4].

To my knowledge no other model experiments exist on spinning active matter. The qualitative confirmation of the phase separation predicted by the Götzel group in [4] is the main strength of the manuscript. The main weakness of this work is arguably its lack of quantitative observations and predictions. While, the particle segregation is clear from the experimental and numerical pictures, virtually no result is convincingly and quantitatively explained about the structure and the dynamics of these spatial heterogeneities. Neither the experiments nor the simulations really advance our understanding of the phenomena first reported in [4]. Numerical simulations could have been employed to go beyond the qualitative observations done on a rather small number of particles. However the simulations are mostly limited to comparison by "visual inspection" (sic). The most quantitative analysis concerns the coarsening dynamics (Fig. 4), and the range of accessible time scales makes it impossible to test the scaling hypothesis reported in [4]. Finally the existence of super-diffusive transport at the interfaces is claimed but not supported by any measurement. (Note also that the edge current discussed in [22] would result in a truly convective dynamics at the interface, as opposed to super diffusion.)

In conclusion, even though the experimental movies and picture are compelling, I feel that this manuscript does not include the necessary amount of quantitative results and new physics to warrant publication in Nature Communications.

More specific comment comments:

-Referring to chiral grains as robot is a strong overstatement.

-Last sentence of the abstract "By mapping rotor motion on a Langevin equation [4] and by comparison with computer simulations, we confirm that our macroscopic system reproduces the dynamics of microswimmers and bacteria." At the very best this statement is unsubstantiated by any convincing reference (note that the physics reported in [28] is very distinct from that discussed in this manuscript).

REPLY to referee 1:

This is an interesting work on segregation/coarsening dynamics of a mixture of robots rotating at different directions. The results evidence analogies with microscopic systems supposing a nice link among active matter at different length scales. The manuscript is well written and supposes a step forward in a field that is raising considerable interest among physicists during the last years. Also, the matching of experimental results with the Langevin simulations reveals that the later can successfully reproduce macroscopic behavior. For all these reasons I think that this work could eventually deserve publication in Nature Communications.

Our reply: We thank the referee for the carefully analysis of our manuscript and recommending our manuscript for eventual publication in Nature Communications. Please find our replies to the concerns in the following.

Nevertheless, I expect something more from a paper in this journal. In particular I think that a systematic analysis of the effect of area filling fraction in the developed patterns (similar to that in ref. 4) is necessary to give the paper the robustness and generality that a journal such as Nature Communications requires.

Our reply: We have added measurements of the stationary state of the system in dependence on the packing fraction (new Figs. 2(h),(i)), a quantitative analysis of the edge currents at the domain interfaces (revised Fig. 5) and provide experimental realizations of the symmetric states (new Figs. 6 (h),(i)). We find the most that the segregation is most pronounced at about 50% filling fraction. Surprisingly, the segregation disappears for larger packing fraction. This can be qualitatively understood from the dynamics of the segregation mechanism (see revised Fig. 3 and corresponding discussion). For a more detailed investigation of the kinetics of the phase transition we would need to study larger systems in order to suppress influences due to finite system size. Since for mechanical reasons the size of the vibrating table cannot be further increased, larger systems can only be achieved by miniaturization of the particles which requires a different 3d printing technology. Further, for significantly larger systems, the initial state of the particles cannot be arranged manually, instead a robotic machine will be used. Such work which is beyond the scope of the present manuscript is currently in progress in our institute,

Other mostly minor issues:

- I cannot see the supplementary videos 4, 5 and 6.
- Fig 1a and 1b are irrelevant for this work.
- A typical simulation trajectory is shown → A typical simulation evolution is shown
- The computation of the average size of domains should be explained with more detail in the Methods section.

Our reply: We re-encoded the videos to improve compatibility. Subfigures (a),(b) from Fig. 1 have been removed to make the focus of the paper clearer. The text has been changed due to the referee's suggestion. The explanation of the calculation of the domain size was added to the Methods section and further details are provided in the Supplementary Information.

We thank the reviewer for the valuable remarks which helped to improve the manuscript. We hope that, after revision, the referee can recommend publication of the manuscript in Nature Communications.

REPLY to referee 2:

‘Rotating Robots Move Collectively and Self-Organize’ by Scholz et al., presents a macroscopic active matter system consisting of an ensemble of 3D-printed robots that rotate and interact sterically. They observe phase-separation into clockwise and anti-clockwise rotators, and characterize the coarsening dynamics of their system and the dynamics of individual robots in the ensemble. Their experimental results are backed up with simulations that quantitatively reproduce the observed dynamics.

Our reply: We thank the referee for the detailed report and constructive criticism. We provide a detailed reply in the following.

This work presents a new example of an active matter system. There are some exciting experimental results, e.g., the dynamics similar to spinodal decomposition and the multiple plateaus at longer times. However, in my opinion, the work should go further in using these results to give insights into non-equilibrium phenomena in general. Without this, the work is unlikely to be of wide general interest to the active-matter community. Thus I believe that, in its current form, this work is more suitable for a specialist journal rather than Nature Communications. With further work it could be suitable for Nature Communications. I expand on this in the following paragraphs.

Our reply: As suggested, we provide several further experimental results in the revision. We performed additional experiments and analysis of our previous results. First, we now show the density dependence of the segregation process, where we find the effect of segregation most pronounced for a density about 50% and a breakdown of segregation at larger density (revised Fig. 2), which can be understood from the segregation mechanism (see revised Fig. 3 and corresponding discussion). We analyze the edge currents quantitatively (see new Fig. 4) and find our observation in agreement with [4]. We would also like to point out that experimental insight into the growth dynamics of active matter system is significantly less frequent than analytical and numerical work. For example in the case of rotors our work is the first to show an experimental realization of the numerically predicted effect.

As the authors state, motility-induced liquid-gas phase separation has already been observed with macroscopic robots [2], [11-13]. The authors here observe a different type of phase separation based on rotor chirality. However, I do not believe that this represents a significant advance over the previous work. If it does, the authors should explain why.

Our reply: In contrast to previous studies, we observe a phase-separation reminiscent of a binary fluid system with high symmetry between both phases. This state cannot appear in a monodisperse system. We believe our experiment is one of the simplest yet non-trivial examples for chiral symmetry breaking in a classical system and goes far beyond the well-known swarming and flocking behavior of macroscopic agents. We added a corresponding remark to the introduction.

The authors claim that the early dynamics of their system is well described by spinodal decomposition. In my opinion, the experimental evidence to back up this claim is not sufficient, consisting of three experimental curves, cropped to only show a particular time window, Fig. 4a-b. One of these curves does not even show the relevant $t^{1/3}$ behaviour, as discussed below. In addition, even if this claim is true, similar results have already been found in simulations of colloidal active matter, and the authors may want to refer to these results, e.g., Stenhammar et al., PRL 111, 145702 (2013). The simulation results on long-time plateaus are interesting, but I could not tell whether the authors have any experimental results on this. Even just from the simulations, it would have been helpful to have some more discussion of these plateaus, e.g., the metastability of the associated states and the lifetime of the plateaus, and how these plateaus are expected to depend on system size (I expect that they may just be due to the small system employed, and for large-scale systems would disappear entirely).

Our reply: We do not believe any experimental curve disagrees qualitatively with the $t^{1/3}$ behaviour. However, we agree our discussion of the results was misleading. First we show the good agreement between experiment and simulation. Similar to experiments on mips, a large ensemble average is impossible in our current experiments (Each experimental run requires assembly of a defined initial state, several hours of continuous measurement and produces a large amount of raw data), we perform a large number of realizations in the simulation. Since in similar simulations in ref [4] it has been shown that the system obeys the correct scaling laws we believe this comparison is correct. We also believe our system poses an interesting problem for analytical treatment such as a derivation of a continuum theory, as mentioned by the referee in the following. We added references ([32]-[34]) to previous analytic and numerical work on the growth dynamics of solid clusters in active matter.

The authors several times suggest other comparisons between their results and previous work on microscopic active and passive systems. However, the details of this comparison are often unclear, e.g., in the concluding paragraph, to what extent do the authors expect that the evolution observed in their experiments agrees with analogous results for microscopic systems, and do the

references cited for this statement confirm their expectations? In general, more engagement with previous work on microscopic active matter would help to widen the interest of this work, e.g., 'Numerous results in the field of active matter have been discovered...' could do with some examples from the literature.

Our reply: It is true that the correspondence to microscopic system needs further experimental work. We mention this in the concluding paragraph and added several references ([19],[39]-[42]) to further support our assumption.

The form of F_i and τ_i should be in the main text, and this should be much clearer and more explicit even if in the SI: SI Eq. 1 presents the force as a scalar quantity, so it is unclear to me whether there is any tangential interaction between colliding cylinders, or whether the interaction is purely along the line of centres; how does the force depend on the overlap between cylinders, and what is ξ ? Also, this model employs a finite Young's modulus, yet no deformation is visible in the movies. Would a strictly rigid model, as employed in [4] give the same phenomenology, or does the modulus have a significant effect?

Our reply: We have added an extended explanation of the force-law we used to the manuscript and supplementary material. As mentioned in the manuscript, ξ is the compression, sometimes also called indentation depth. Tangential forces between colliding discs are not considered. The force-law used in [4] was a Weeks-Chandler-Anderson potential, so this was also not strictly rigid. Due to the high Young's modulus the deformation is only microscopic in our system (the particles are made from a rigid plastic). Note, that we observe no qualitative differences when using other force laws (such as spherical Hertz contact or WCA), but wanted to use a realistic established model for our granular particles.

The definition of the random forces and torques is confusing after Eq. 1-2. I believe that this is a 2d model, so that there should only be two spatial dimensions for the random force and 1 spatial dimension for the random torque. Instead, both of these have three dimensions specified.

Our reply: Correct. We have changed the definitions accordingly.

In Fig. 2g, arrows on the red/blue trajectories would be helpful. Also, the caption currently reads 'clockwise diffusion of clockwise-rotating particles...' etc., but the motion is non-diffusive.

Our reply: We modified the figure, added arrows to indicate the direction of the particle motion changed the corresponding caption.

The units of g should be $ms^{-1/3}$ not $ms^{1/3}$ and the parameter w should be on the order of one particle diameter, not just greater than one particle diameter, for the order-of-magnitude calculation of g to make sense.

Our reply: Correct. We have changed the manuscript accordingly.

Some explicit cross-reference to the SI would be useful, e.g., when discussing the mass and moment of inertia.

Our reply: We have added a cross-reference to the SI.

One of the red curves in Fig. 4a-b appears to show a plateau at longer times, but the authors claim that all these curves are in the intermediate-time regime where no plateaus should occur. Are these plateaus a significant, reproducible feature? More experiments are needed to demonstrate this one way or the other. Why do the authors not show the long-time dynamics courses for the experiments as well as for the simulations?

Our reply: Plateaus can also occur at smaller times, due to the finiteness of the system (see Fig. 6(a),(b)). However this is rate at small times and plateaus become more stable and frequent for later times. We have added two snapshots from the experiments, where we observe high-symmetry states in current experiments(Fig. 6(h),(i)). Unfortunately the formation of these plateaus can also take a long time beyond the time scales where our experiments is stable against gravitational drift. We have added a section to the Supplementary Information where we discuss this aspect. A systematic experimental analysis is a current project in our group, but will require a significant decrease of particle size and automated particle placement, otherwise a significant number of ensembles cannot be obtained in a reasonable time.

In what way is the phase separation reminiscent of droplet formation, and why are these connections surprising? A reference to a work on droplet formation would be helpful here.

Our reply: We observe that interfaces typically form with a certain curvature and hit the boundary at an angle of 90° between the two phases. We believe this is similar to contact-angles between immiscible fluid mixtures at boundaries. However, we know

believe the term 'droplet formation' is misleading. We have added a more detailed explanation and corresponding references ([36],[37]) to the manuscript.

We wish to thank the referee again for the very careful reading and the helpful report. We hope that after revision the referee can recommend the manuscript for publication in Nature Communications.

REPLY to referee 3:

The authors report the first experimental investigation of the collective dynamics of spinning active particles, a subject which has recently triggered a surge of research in active-matter physics. Using suitably designed grains shaken by a vibrating stage, they demonstrate the segregation of particles spinning in opposite direction and claim the observation of subdiffusive motion at the domain interfaces. These observations are consistently reproduced by a numerical simulations similar to that first introduced in [4].

Our reply: We thank the referee for the helpful criticism and for raising important questions. Please find our detailed reply in the following.

To my knowledge no other model experiments exists on spinning active matter. The qualitative confirmation of the phase separation predicted by the Glotzer group in [4] is the main strength of the manuscript. The main weakness of this work is arguably its lack of quantitative observations and predictions. While, the particle segregation is clear from the experimental and numerical pictures, virtually no result is convincingly and quantitatively explained about the structure and the dynamics of these spatial heterogeneities. Neither the experiments nor the simulations really advance our understanding of the phenomena first reported in [4]. Numerical simulations could have been employed to go beyond the qualitative observations done on a rather small number of particles. However the simulations are mostly limited to comparison by visual inspection (sic). The most quantitative analysis concerns the coarsening dynamics (Fig. 4), and the range of accessible time scales makes it impossible to test the scaling hypothesis reported in [4].

Our reply: We agree the experimental confirmation of an effect, previously only observed in simulations provides a significant advance. To support the underlying physical principles we have added a detailed experimental investigation in dependence of the area filling fraction(see revised Fig. 2). Additionally we have added trajectories of colliding particles to illustrate the mechanism of separation(see Fig. 3). We believe that our system can eventually lead to a more quantitative test of the scaling hypothesis. However this requires significant advances on the experimental level, which we believe, is only feasible as a collaborative effort of the community, which hopefully will be possible in future systems. We have added a discussion on this problem to the outlook of the paper.

Finally the existence of super-diffusive transport at the interfaces is claimed but not supported by any measurement. (Note also that the edge current discussed in [22] would result in a truly convective dynamics at the interface, as opposed to super diffusion.)

Our reply: We have added a quantitative discussion of the directed transport at the interfaces, in analogy to [4]. Due to the limited size of the system however, it is not possible to quantitatively distinguish both scenarios. However, we can clearly observe quantitatively the increased and directed (ballistic) displacement at the boundary. We have added Fig. 4 and a corresponding paragraph to the manuscript to explain this effect.

In conclusion, even though the experimental movies and picture are compelling, I feel that this manuscript does not include the necessary amount of quantitative results and new physics to warrant publication in Nature Communications.

Our reply: We hope that, after significant revision, the referee can recommend the manuscript for publication in Nature Communications.

More specific comment comments: Referring to chiral grains as robot is a strong overstatement;

Our reply: It is true, that our particles are very simplistic machines. However we believe that our particles, fulfill all requirements of automatized and externally driven independent systems. Similar designations are typical in literature, such as the term "living crystals" for collectives of microswimmers.

Last sentence of the abstract By mapping rotor motion on a Langevin equation [4] and by comparison with computer simulations, we confirm that our macroscopic system reproduces the dynamics of microswimmers and bacteria. At the very best this statement is unsubstantiated by any convincing reference (note that the physics reported in [28] is very distinct from that discussed in this manuscript).

Our reply: Correct. Our statement was misleading. While we believe the assumption that a realization on the microscopic scale is reasonable, an experimental confirmation is missing and requires further experimental effort. We have changed the sentence to clarify our statement.

We wish to thank the referee again for the very careful reading and the helpful report. We hope that after our revision the referee can recommend the manuscript for publication.

Reviewers' comments:

Reviewer #1 (Remarks to the Author):

Dear Editor,

I have read the revised version of the manuscript "Rotating Robots Move Collectively and Self-Organize". The authors have done an important effort to answer all my questions (and most of the other referees) including new experiments and measurements.

In its present version, I believe that the manuscript fulfills Nature Communications criteria of novelty and impact. Therefore, I support its publication as it is.

I only recommend the authors rewriting the paragraph starting by: "We quantify the edge currents...", in particular, the second part of it. The three conclusions obtained from figures a) and b) are a bit mixed. Also, there is a typo in the explanation of the first observation (I think there is an "are" missing).

Sincerely yours,

The referee.

Reviewer #2 (Remarks to the Author):

I am satisfied with the authors' response to almost all of my comments and I am satisfied that this publication is now almost suitable for publication in Nat. Comm.. In particular, the further experiments on different area fractions seriously enhance the manuscript, and the discussion and connection to previous work is now much clearer.

However, I am still concerned about Fig. 6a-b. The point is that there are only three experimental curves, and one of these shows a plateau that is not mentioned in the text, as far as I can tell. Am I supposed to ignore the plateau when considering whether the curves show a $t^{1/3}$ behaviour, given that the curve with a plateau shows an average $\sim t^{1/6}$ behaviour overall? Why should I do this if there are only two other curves? At the very least, this requires some discussion of the plateau (which, by the way, appears to be consistent with the lowest plateaus (f,g) in Fig. 6c). Ideally, a few

more experimental curves would make the manuscript more convincing. From the authors' description of their experiments, which they say take hours (as opposed to days or weeks), I do not believe this is an unreasonable request, but I agree that it is not 100% necessary, so I leave this to the editor's discretion.

Despite the authors' statement in their reply, the parameter w is still quoted as being > 15 mm. As I stated previously, this does not support the authors' order of magnitude estimate for g . The parameter w must instead be of order 15 mm for the estimate of g to follow.

The new experiments also raise a couple of concerns: on the basis of the errorbars in Fig. 2i, there is no statistically significant evidence for a peak at 50%; rather, the data is consistent with a plateau from 30% up to 55%. Similarly, I do not see significant evidence for hysteresis. If the authors toned down these claims, that would be sufficient.

REPLY to referee 1:

I have read the revised version of the manuscript “Rotating Robots Move Collectively and Self-Organize”. The authors have done an important effort to answer all my questions (and most of the other referees) including new experiments and measurements.

In its present version, I believe that the manuscript fulfills Nature Communications criteria of novelty and impact. Therefore, I support its publication as it is.

Our reply: We thank the referee for the report and the clear recommendation to publish the manuscript.

I only recommend the authors rewriting the paragraph starting by: “We quantify the edge currents”, in particular, the second part of it. The three conclusions obtained from figures a) and b) are a bit mixed. Also, there is a typo in the explanation of the first observation (I think there is an “are” missing).

Our reply: We modified the manuscript as suggested by the referee.

REPLY to referee 2:

I am satisfied with the authors' response to almost all of my comments and I am satisfied that this publication is now almost suitable for publication in Nat. Comm.. In particular, the further experiments on different area fractions seriously enhance the manuscript, and the discussion and connection to previous work is now much clearer.

Our reply: We thank the referee for the kind assessment of our revised manuscript.

However, I am still concerned about Fig. 6a-b. The point is that there are only three experimental curves, and one of these shows a plateau that is not mentioned in the text, as far as I can tell. Am I supposed to ignore the plateau when considering whether the curves show a $t^{1/3}$ behaviour, given that the curve with a plateau shows an average $\sim t^{1/6}$ behaviour overall? Why should I do this if there are only two other curves? At the very least, this requires some discussion of the plateau (which, by the way, appears to be consistent with the lowest plateaus (f,g) in Fig. 6c). Ideally, a few more experimental curves would make the manuscript more convincing. From the authors' description of their experiments, which they say take hours (as opposed to days or weeks), I do not believe this is an unreasonable request, but I agree that it is not 100% necessary, so I leave this to the editor's discretion.

Our reply: Following the suggestion of the referee, we added four new experimental runs to Fig. 6a,b. The new experimental data also consistently falls in the interval indicated in Fig. 6a,b by two black lines and therefore is in qualitative agreement with the $t^{1/3}$ claim. Nevertheless, even when new and old data is combined, we feel we cannot clearly exclude all other competing scenarios such as the one mentioned by the referee. Here, we would like to stress that a precise determination of the value of the growth exponent has not been an aim of our manuscript but is merely discussed for comparison with prior works. The finding of phase separation by itself is the main novelty here. We added and modified the following three sections on pages 3-5:

"Variations among experimental runs appear to be slightly larger than variations among the simulation runs, possibly caused by mass polydispersity and shape imperfections of our 3d-printed rotors. Still, overall good agreement in the average coarsening behavior is observed. This demonstrates that the system of Langevin equations fitted to single-particle behavior also reproduces collective dynamics well."

"Nevertheless, the comparison to a power-law is only qualitative and is affected by the confinement condition. Simulations of large systems of rotors without confinement have already shown with high accuracy that the Langevin model used here leads to spinodal decomposition with exponent $1/3$ [25]. Larger setups than ours are necessary to accurately determine the growth exponent in experiment."

"We note that in experiment the fully segregated symmetrical state is found only for small system sizes (Fig. 2(h)). For the large system of Fig. 6 the time to reach the final state is so long that gravitational drift at the boundaries eventually leads to asymmetry (see Supplementary Information)."

Despite the authors' statement in their reply, the parameter w is still quoted as being > 15 mm. As I stated previously, this does not support the authors' order of magnitude estimate for g . The parameter w must instead be of order 15 mm for the estimate of g to follow.

Our reply: We corrected the text following the referee's comment.

The new experiments also raise a couple of concerns: on the basis of the errorbars in Fig. 2i, there is no statistically significant evidence for a peak at 50%; rather, the data is consistent with a plateau from 30% up to 55%. Similarly, I do not see significant evidence for hysteresis. If the authors toned down these claims, that would be sufficient.

Our reply: We agree that our claims of the appearance of a peak and hysteresis are not sufficiently supported by the experimental data. We changed the caption to Fig. 2i and the manuscript text correspondingly:

"For $0.3 \leq \phi \lesssim 0.55$, this quantity is independent of packing fraction indicated by a plateau."

"A plateau is found in the range $\phi \lesssim 0.55$ before the system starts mixing again at higher packing fraction (Fig. 2(i))."

We wish to thank the referee again for the helpful report. We hope that after this revision the referee can recommend the manuscript for publication in Nature Communications.

REVIEWERS' COMMENTS:

Reviewer #2 (Remarks to the Author):

I am satisfied with the authors' responses to all my comments, and am happy to recommend publication in Nature Communications.

[redacted]